# Genome-Wide SNPs and InDels Characteristics of Three Chinese Cattle Breeds

**DOI:** 10.3390/ani9090596

**Published:** 2019-08-22

**Authors:** Fengwei Zhang, Kaixing Qu, Ningbo Chen, Quratulain Hanif, Yutang Jia, Yongzhen Huang, Ruihua Dang, Jicai Zhang, Xianyong Lan, Hong Chen, Bizhi Huang, Chuzhao Lei

**Affiliations:** 1Key Laboratory of Animal Genetics, Breeding and Reproduction of Shaanxi Province, College of Animal Science and Technology, Northwest A&F University, Yangling 712100, China; 2Yunnan Academy of Grassland and Animal Science, Kunming 650212, China; 3National Institute for Biotechnology and Genetic Engineering, Pakistan Institute of Engineering and Applied Sciences, Faisalabad 577, Pakistan; 4Institute of Animal Science and Veterinary Medicine, Anhui Academy of Agriculture Science, Hefei 230001, China

**Keywords:** cattle, SNP, InDel, whole-genome resequencing

## Abstract

**Simple Summary:**

Whole-genome resequencing is an important tool to reveal the in-depth genomic characteristics of a genome. Adaptability traits are key to the survival of the south Chinese zebu cattle. However, the potential genetic information behind these remarkable traits still remains uncertain and needs to be addressed. In the current study, we utilized a total of 15 local south Chinese cattle samples (Leiqiong (LQ), Wannan (WN), Wenshan (WS)) from one of our previous studies mapped to the old reference genome (Btau_5.0.1) and remapped them to the latest reference genome (ARS-UCD1.2) to explore potential single nucleotide polymorphisms (SNPs) and insertions-deletions (InDels) responsible for some important immune related traits. The present study emphasizes and illustrates the genetic diversity, extending our previous study. The InDel annotation show that WS cattle had more enriched genes associated with immune functions than the other two breeds. Our findings provide valuable resources for further investigation of the functions of SNP- and InDels-related genes and help to determine the molecular basis of adaptive mutations in Chinese zebu cattle.

**Abstract:**

We report genome characterization of three native Chinese cattle breeds discovering ~34.3 M SNPs and ~3.8 M InDels using whole genome resequencing. On average, 10.4 M SNPs were shared amongst the three cattle breeds, whereas, 3.0 M, 4.9 M and 5.8 M were specific to LQ, WN and WS breeds, respectively. Gene ontology (GO)analysis revealed four immune response-related GO terms were over represented in all samples, while two immune signaling pathways were significantly over-represented in WS cattle. Altogether, we found immune related genes (*PGLYRP2*, *ROMO1*, *FYB2*, *CD46*, *TSC1*) in the three cattle breeds. Our study provides insights into the genetic basis of Chinese indicine adaptation to the tropic and subtropical environment, and provides a valuable resource for further investigations of genetic characteristics of the three breeds.

## 1. Introduction

The domestication of cattle was one of the most significant happenings around 10,000 years ago, assisting mankind with increasing meat, milk and leather supplies, throughout the world [1]. Domestic cattle adapted to various diverse environmental conditions during the natural selection, such as Brahman cattle adapted to harsh tropical conditions whereas Yakutian cattle adapted to the subarctic environment [2,3]. The genetic selection of cattle, e.g., Holstein and Beef master, led to the production of higher milk/meat than local cattle breeds [4,5]. At present, more than 800 cattle breeds have been identified in the world [6], and these cattle breeds constitute an important world heritage and a unique genome resources.

China has a vast territory, and harbors 53 domestic cattle breeds [7]. According to the previous studies [8,9,10], Chinese cattle can be geographically classified into three categories: the northern group distributed in the north of China (*Bos taurus*), the central group located in the middle and lower areas of the Yellow River and the Huaihe River (a mixture of *Bos taurus* and *Bos indicus*) and the southern group in the south of China (*Bos indicus*).

Considerable progress has been made through high-throughput sequencing to obtain cattle whole-genome sequences, which offer extremely promising approaches for screening the molecular targets of disease and resistance. Recently, a large number of genome-wide resequencing data for different cattle breeds have been published, including 11 indigenous Pakistani cattle breeds [11], Brahman cattle [12] and Indian cattle [13,14]. These studies have enriched the sequencing data of different breeds and also provided information for studying the genetic diversity of different domestic cattle breeds.

Until now, Chinese indicine cattle has been lacking behind in its detailed gene characterization. In the present study, we performed whole-genome resequencing of 15 individuals of three indigenous indicine breeds, including Wenshan cattle (WS, n = 7), Wannan cattle (WN, n = 5) and Leiqiong (LQ, n = 3) cattle. The aim of our study is to provide a valuable resource for further investigations of the genetic mechanisms underlying traits of interest in Chinese indicine cattle.

## 2. Materials and Methods

### 2.1. Whole-Genome Data

We used whole-genome data of three Chinese cattle breeds (n = 15) from our previous study (Table 1) [13]. DNA was extracted from the ear tissues of each individual using the standard phenol-chloroform method. Two libraries with insert sizes of 500 bp were constructed for each individual and sequenced using the HiSeq 2000 platform (Illumina, Beijing, China).

### 2.2. Read Mapping and SNP Calling

The reads were mapped to the latest reference genome (ARS-UCD1.2, *Bos taurus*, Breed Hereford) using BWA-mem [15], according to the default parameters. The Genome Analysis Toolkit (GATK, version 3.8) was further employed for SNP calling, followed by mark Duplicate by Picard [16]. GATK, “variant Filtration” was implemented for all SNPs as follows: (1) variant confidence/quality by depth (QD) <2; (2) RMS mapping quality (MQ) >40.0; (3) Phred-scaled *p*-value using Fisher’s exact test to detect strand bias (FS) <60; (4) z-score from the Wilcoxon rank sum test of Alt vs. Ref read position bias (ReadPosRankSum) >−8; (5) z-score form the Wilcoxon rank sum test of Alt vs. Ref read mapping qualities (MQRankSum) >12.5; and (6) variants with SOR (symmetric odds ratio of 2 × 2 contingency table to detect strand bias) >3.0.

### 2.3. Identification of InDels

For all the 15 cattle samples, the InDels were extracted by GATK [16] in the “variant Filtration” within 1 bp to 30 bp window with the following parameters: (1) variant confidence/quality by depth (QD) <2; (2) phred-scaled *p*-value using Fisher’s exact test to detect strand bias (FS) >200; (3) z-score from the Wilcoxon rank sum test of Alt vs. Ref read position bias (ReadPosRankSum) <−20; (4) Likelihood-based test for the consanguinity among samples (InbreedingCoeff) <−0.8.

### 2.4. Variant Functional Annotation and GO Enrichment

The SNP/InDels were classified as HIGH, MODERATE, LOW and MODIFIER as per their functionality using SnpEff [17]. Only HIGH and MODERATE impact SNPs/InDels were kept for further analysis, including SNPs with (HIGH: protein truncation or triggering loss/gain of function; MODERATE: missense variant and splice variant that could change protein effectiveness); whereas the InDels include (HIGH: frameshift variant or splice donor variant; MODERATE: disruptive inframe insertion) [11]. For all the individuals, the variants were filtered as >5 SNPs/gene, whereas >5 InDels/gene were identified per breed [3]. The SnpEff functional class vocabulary assigned to both SNPs and InDels were Untranslated Region(UTR) variant include 3 prime UTR and 5 prime UTR, Downstream and upstream gene variant, intergenic region, intragenic, intron, non-coding transcript exon and non-coding transcript variant, splice variant (splice acceptor, splice donor and splice region), start lost, stop gained, stop lost and stop retained. Functional classes exclusively used for InDels were conserved in-frame deletion and insertion, disruptive in-frame deletion and insertion, bidirectional gene fusion, frameshift, conservative-disruptive inframe InDel, whereas missense, initiator codon and synonymous were only assigned to SNPs.

The Gene Ontology online tool (http://geneontology.org/) was used to identify over-represented biological process using the filtered variants (*Bos taurus* reference genome build 9913, Released 2019-4-17) [18,19]. Fisher’s exact test with calculate false discovery rate testing was executed and a *p*-value of less than 0.05 was chosen as an inclusion criterion for functional categories.

## 3. Results

### 3.1. Read Alignment

Three breeds (LQ, n = 3; WN, n = 5; WS, n = 7) were distributed in Haikou city, Guangdong province; Jinde county, Anhui province; and Guangnan county, Yunnan province from southern China. The average genome coverage to the reference genome was 99.17% (ranging from 97.88% to 99.56%) with an elevated in-depth mapping coverage of 11.86 folds (ranging from 6.53 to 21.36) and 6.9% duplication rate.

### 3.2. Identification of SNPs and InDels

After quality filtering, ~34.3 million SNPs were identified across all 15 samples, in relation to the latest taurine reference genome (GCF_002263795.1) (Appendix A). The SNPs density was also detected to be approximately 13791 SNPs per million bp (MB) in all samples, whereas 7703, 9275 and 9412 SNPs/MB were calculated in LQ, WN, WS, respectively. With the transition (Ts) versus transversion (Tv) ratio ranges from 2.375 to 2.390 (Appendix A). Compared with NCBI dbSNP bovine, a total of 9.33 million novel SNPs was identified, whereas, approximately 14.8, 17.8 and 18.2 million SNPs were previously identified and annotated in LQ, WN and WS cattle (Table 2, Appendix A). The number of InDels (mostly ≤3 bp) was 2,153,542; 2,586,758; and 2,471,063 in LQ, WN and WS cattle, respectively (Figure 1, Appendix A). Among all InDels, 2,294,379 (59.82%) are deletions, whereas 1,453,791 (58.83%),; 1,508,944 (58.33%); and 1,257,000 (58.37%) are breedwise deletions of LQ, WN and WS cattle, respectively (Appendix A).

In this study, 10,446,053 SNPs were shared amongst the three breeds, whereas, 3,046,955 (15.89%), 4861786 (21.05%) and 5,773,919 (24.64%) were specific to LQ, WN and WS cattle. Furthermore, 1,065,517 InDels were shared among all samples, whereas, 432,710; 662,590; and 668,308 number of InDels were private to LQ, WN and WS breeds, respectively (Figure 2).

### 3.3. Functional Annotation of SNPs and InDels

Functional classes of the SNPs identified in this study are shown in Table 2. The numbers of functionally annotated SNPs were slightly higher than those of the detected SNPs, the reason for which lies with the possible presence of multiple annotations to a single SNPs. As expected, most of the SNPs discovered in this study map to intergenic 24,610,698 (9.01%) and intronic regions 151,807,389 (55.61%) and are potentially neutral; 8,664,911 (3.17%) and 8,403,001 (3.08%) SNPs are positioned in a 1000 bp downstream and upstream regions from the genes set, respectively. These SNPs classified as HIGH and MODERATE impact, include 300,763 (0.11%) missense mutation and 119,959 (0.04%) splice mutation, whereas many other SNPs, such as stop-gain, start-loss and stop-loss (2284, 465 and 400, respectively) were also detected in the current study.

Functional annotation depicted all samples, the location of 2,037,409 (6.72%) InDels in intergenic and 16,689,877 (55.07%) in intronic regions, a large number of InDels exist in untranslated regions (103,089 InDels in 3 prime UTR and 19,146 InDel in 5 prime UTR). A total of 4018 and 2688 InDels were annotated as Disruptive and Conservative inframe InDels respectively. Additionally, 12,894 InDels induced frameshift mutations, 14655 InDels affected splice-sites (splice regions, splice donor and splice acceptor variants), and two InDels resulted in premature stop codons. At the breed level, most variants were identified in intergenic region, intragenic region and intron, while a small number of variants had a bearing on protein translation, such as, disruptive inframe InDel, split variants and frameshift variants (Table 3).

### 3.4. GO Analysis of the SNPs and InDels

The GO enrichment analysis of 1260 filtered genes identified 58 significant GO terms in all 15 cattle (Table 4, Appendix A). The GO enrichment analysis revealed four GO categories associated with sensory perception processes (GO:0007608~sensory perception of smell; GO:0050911~detection of chemical stimulus involved in sensory perception of smell; GO:0007606~sensory perception of chemical stimulus; GO:0050907~detection of chemical stimulus involved in sensory perception). Five GO biological processes related to metabolic process (GO:0006956~complement activation; GO:0006959~humoral immune response; GO:0006958~complement activation, classical pathway; GO:0002253~activation of immune response) were found. Four immune responses related GO terms were also identified (GO:0006956~complement activation; GO:0006959~humoral immune response; GO:0006958~complement activation, classical pathway; GO:0002253~activation of immune response).

GO enrichment analysis of 1207, 1493, and 1937 filtered genes associated with frameshift InDel in LQ, WN and WS cattle, respectively. The GO enrichment analysis revealed16, 24, and 19 GO terms to be associated with biological processes in LQ, WN and WS cattle, respectively (Appendix A). The GO enrichment analysis showed 14 significantly enriched GO terms, shared by the three cattle breeds. Three noticeably enriched GO terms (GO:0023052~signaling; GO:0007165~signal transduction; GO:0007154~cell communication) were shared by WS and WN (Appendix A). Amongst three breeds, two GO terms related to immune response (GO:0002252~immune effector process; GO:0006956~complement activation) were enriched in WS cattle, alone. On the other hand, in the WN breed, a large number of genes were significantly associated with metabolic process functions, such as GO:0051171~regulation of nitrogen compound metabolic process; GO:0080090~regulation of primary metabolic process; GO:0019222~regulation of metabolic process; GO:0060255~regulation of macromolecule metabolic process; GO:0031323~regulation of cellular metabolic process; GO:0008152~metabolic process; GO:0010605~negative regulation of macromolecule metabolic process. In LQ cattle, two of the GO terms associated with L-lysine transport were enriched.

## 4. Discussion

We herein carefully examined the whole-genome sequences of three aboriginal breeds in China. The TS/TV ratio could evaluate the value of the resequencing error, which was used to assess the quality of the SNPs. In our study, TS/TV ratio ranges from 2.375 to 2.39, which is similar to previous studies (Appendix A) [20,21]. The homozygous and heterozygous SNP ratio in each breed indicate the normalization of the population structure.

The three breeds we studied are native to southern China regions with a subtropical climate. Compared with *Bos taurus*, the indicine has the presence of a hump, loose skin and shorter and thinner hair, and they all have the characteristics of adapt to the hot climate and resisting diseases [22]. Breed-shared SNP could possibly be helpful in further research common function or phenotype of *Bos indicus*. From all 29 autosomes, we identified more than 34.3 million SNPs and ~3.84 million InDels (Table 2 and Table 3), of which approximately 30.43% and 27.78% shared in all 15 cattle. We also found that 3.05 million (15.89%) in LQ, 4.86 million (21.05%) in WN and 5,773,919 (24.64%) in WS were private SNPs in our SNPs set (Figure 2). These breed specific SNPs provided conditions for breed characterization of the further research.

In our study, WS has the greatest number of SNPs, and WN has the greatest number of InDels. Interestingly, LQ has the minimum number of in both SNPs and InDels (Table 2 and Table 3). The lengths of the InDels ranged from −30 bp (deletion) to 30 bp (insertion). However, the small InDels (≥3 bp) account for 83.07% of the total InDels, which is comparable to the previous results [11,21].

Our survey of three geographical distinct indicine cattle breeds (LQ, WS, WN) showed that each of them has similar characteristic. GO analysis revealed that a lot of immune-related gene were shared by all samples. Among them, the *PGLYRP2* gene is an important gene involved in bacterial infection immune response [23]. Studies have shown that this gene is related to somatic cells count in milk [24] and immune response to advantageous and harmful gut bacteria [25]. These results suggest that the *PGLYRP2* gene may be associated with bovine gut bacteria and milk quality. *ROMO1* gene encodes a mitochondrial membrane protein that has the effect on increasing intracellular reactive oxygen species [26]. Recent studies have shown that the *ROMO1* gene product are highly expressed in cancer cells and triggers sustained inflammatory response [27]. In addition, *PTNR22* are associated with immune diseases [28,29]. *STK11* gene is a tumor suppressor gene [30]. *FYB2* gene, also known as *ARAP* gene, encodes a T cell adaptor protein mediating cell adhesion [31]. These immunity-related genes were shared between three breeds of cattle. In the present study, three breeds enriched by common immune enrichment pathway, suggests that Chinese indicine may have some common mechanisms towards adaption to the environment in Southern China. In InDels, the immune response-related genes were only enriched in WS cattle. Among them, CD46, a type 1 transmembrane glycoprotein, whose main function is to regulate complement activation [32]. We found 55 InDels related to the *CD46* gene in WS cattle, suggesting that WS cattle may have well antiviral infection characteristics [33,34]. EMP2 has been shown to promote angiogenesis in vitro and in vivo [35]. *TSC1* gene encodes the growth inhibitory protein hamartin and increase gene expression contributing to cardiovascular health [36]. These genetic mutations may be important factors in WS cattle adaption to the local environment. In addition, we identified some InDels related to metabolism in WN cattle, and other genes related to lysine transport in the LQ breed.

## 5. Conclusions

Our study used resequencing data from three cattle breeds to provide detailed genomic information, including SNPs and InDels. Amongst them, WS cattle contained the greatest number of SNPs, which might have resulted in parallel to the maximum number of WS cattle in the current study. However, WN cattle dominated in terms of InDels. The breed-specific genetic variants are crucial for maintaining herds’ genetic diversity and development of its breeding strategies. In our present study, a total of ~3.04 M, ~4.86 M and ~5.77 M SNPs were identified specific to LQ, WN and WS cattle, while, the number of InDels ranged from ~0.432 M, ~0.662 M and ~0.668 M respectively. The Gene Ontology of SNPs enriched immune pathways, revealing *PGLYRP2*, *ROMO1*, *PTNR22*, *STK11* and *FYB2* genes. InDels on the other hand, depicted over representation of immune related GO terms in WS, while L-lysine transport and the metabolism showed over-representation only in LQ and WN respectively. The current study unveils the genetic characteristics of three important southern Chinese zebu cattle breeds, providing the genome resources for further study.

## Figures and Tables

**Figure 1 animals-09-00596-f001:**
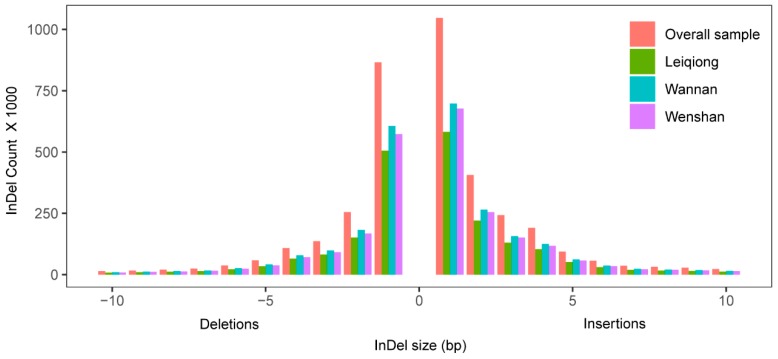
Distribution of insertions and deletions length in all InDels.

**Figure 2 animals-09-00596-f002:**
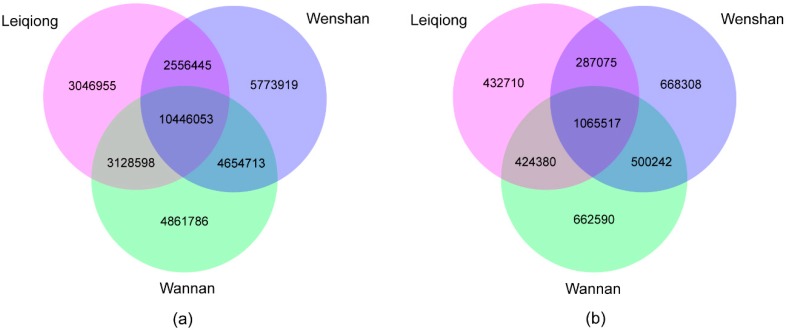
Venn diagram describes overlap and unique SNPs/InDels between the three breeds (LQ, WN, WS). The numbers show specific SNPs/InDels for each breed or overlapping SNPs/InDels between any two breeds or among three breeds. (**a**) The identified shared and specific SNPs for each breed. (**b**) The identified shared and specific InDels for each breed.

**Table 1 animals-09-00596-t001:** Summary of sequencing and mapping results in 15 samples.

Breed	Sample ID	SRR ID	Total Reads	Aligned Reads Rate (%)	Duplication Rate (%)	Average Read Depth
WS	WS1	SRR6024561	132599505	131966081 (99.52%)	5.24%	6.7367×
	WS2	SRR6024562	210348316	208908938 (99.32%)	7.57%	10.9981×
	WS3	SRR6024569	213171893	209992479 (98.51%)	7.56%	11.0636×
	WS4	SRR6024575	220677796	219684659 (99.55%)	7.58%	10.6536×
	WS5	SRR6024576	185678168	184843226 (99.55%)	6.60%	9.3981×
	WS6	SRR6024577	220875686	219906218 (99.56%)	7.17%	11.5984×
	WS7	SRR6024578	124034762	123493668 (99.56%)	5.40%	6.5332×
WN	WN4	SRR5507199	453240314	443641290 (97.88%)	10.12%	22.9631×
	WN8	SRR5507198	453327732	448362891 (98.9%)	9.91%	23.1603×
	WN9	SRR5507195	189321051	187411280 (98.99%)	5.63%	9.6483×
	WN10	SRR5507196	203146450	201321637 (99.1%)	6.00%	10.1569×
	WN11	SRR5507197	229615463	228096174 (99.34%)	6.33%	11.7816×
LQ	LQ5	SRR5507190	229615463	228096174 (99.34%)	6.33%	11.5007×
	LQ12	SRR5507189	219526027	217652138 (99.15%)	6.14%	11.1222×
	LQ15	SRR5507188	208719494	207367555 (99.35%)	5.91%	10.6481×

**Table 2 animals-09-00596-t002:** Functional annotation of the detected SNP variants in three cattle breeds.

Fields	LQ	WN	WS
Total number	19,178,051	23,091,150	23,431,130
3 prime UTR	390,813	481,642	500,217
5 prime UTR	110,542	137,007	146,969
Downstream gene	4,673,996	5,721,422	5,884,594
Initiator codon	25	31	39
Intergenic region	10,333,593	12,409,887	12,561,984
Intragenic	13,622,957	16,715,251	17,111,225
Intron	85,061,834	100,821,955	105,226,537
Missense	138,168	173,641	212,294
Non coding transcript exon	172,858	212,090	223,022
Non coding transcript	32,436,546	39,244,542	39,287,003
Splice acceptor	717	900	887
Splice donor	857	1152	1129
Splice region	59,437	74,604	79,809
Start lost	246	273	329
Stop gained	1108	1253	1476
Stop lost	211	273	263
Stop retained	221	246	258
Synonymous	278,880	348,230	528,134
Upstream gene	4,538,595	5,552,247	5,651,124
Novel	4,354,976	5,288,934	5,226,594
Known	14,823,074	17,802,215	18,204,535

**Table 3 animals-09-00596-t003:** Distribution of SnpEff annotation InDel variants in three cattle breeds.

Fields	LQ	WN	WS
Total number	2,153,542	2,586,758	2,471,063
3 prime UTR	54,892	67,465	65,837
5 prime UTR	9904	12,229	12,421
Bidirectional gene fusion	80	83	139
Conservative inframe deletion	561	814	1021
Conservative inframe insertion	434	546	781
Disruptive inframe deletion	1039	1292	1735
Disruptive inframe insertion	483	583	955
Downstream gene	585,215	719,779	693,994
Frameshift variant	4168	5516	9559
Gene fusion	198	178	232
Intergenic region	1152,941	1,382,145	1,318,068
Intragenic	1,485,117	1,817,239	1,775,709
Intron	9,407,833	11,108,642	10,992,364
Non coding transcript exon	17,425	21,355	21,201
Non coding transcript	3,688,414	4,441,858	4,170,309
Splice acceptor	384	451	471
Splice donor	368	412	477
Splice region	6396	7787	8542
Start lost	58	64	54
Stop gained	57	73	131
Stop lost	32	73	61
Upstream gene	54,6024	666,968	642,273

**Table 4 animals-09-00596-t004:** Gene Ontology (GO) reveals three cattle sharing high or moderate effects >5 SNPs.

GO Biological Process Complete	Count	Fold Enrichment	FDR
Smell			
Sensory perception of smell (GO:0007608)	104	3.05	2.78 × 10^−18^
Detection of chemical stimulus involved in sensory perception of smell (GO:0050911)	101	3.03	5.15 × 10^−18^
Sensory perception of chemical stimulus (GO:0007606)	104	2.96	4.27 × 10^−18^
Detection of chemical stimulus involved in sensory perception (GO:0050907)	101	2.99	6.22 × 10^−18^
Immune responses			
Complement activation (GO:0006956)	9	7.5	3.92 × 10^−3^
Humoral immune response (GO:0006959)	11	4.72	1.62 × 10^−2^
Complement activation, classical pathway (GO:0006958)	6	9.21	3.29 × 10^−2^
Activation of immune response (GO:0002253)	16	3.01	4.58 × 10^−2^
Metabolic process			
Metabolic process (GO:0008152)	114	0.57	2.55 × 10^−10^
Organic substance metabolic process (GO:0071704)	98	0.56	2.16 × 10^−9^
Cellular metabolic process (GO:0044237)	87	0.54	2.92 × 10^−9^
Primary metabolic process (GO:0044238)	95	0.57	5.67 × 10^−8^
Organonitrogen compound metabolic process (GO:1901564)	64	0.55	4.18 × 10^−5^

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
