# Peer review of "Genome-Wide SNPs and InDels Characteristics of Three Chinese Cattle Breeds"

_animals, 2019, doi:10.3390/ani9090596_

Round 1
Reviewer 1 Report
Zhang et al. reported the analysis of polymorphisms present in three Chinese Bos indicus breeds using available genomic sequences. They identified a total of 37 million SNPs and 4 million indels. They showed enrichment in immune infection-related GO terms in the most polymorphic genes compared to Bos Taurus genome. The manuscript is well written and contains only a few typos. Nevertheless, the material and method section needs to be clarified in order to specify what was done in the present study in comparison to reference 13 [Chen et al. 2018]. Besides, the filtering performed on the genes/polymorphisms needs to be clarified and the reference used in the gene ontology analysis needs to be specified. Below a more detailed list of suggested improvements before publications of the manuscript.
Abstracts
- L21. I suggest avoiding abbreviations in the abstract to ease reading (WS, LG and WN) and indicating Latin name of the species analyzed (Bos indicus).
- L19. “The south Chinese indicine (Zebu) has been blessed with remarkable adaptability traits but unfortunately lacks the adequate genetic information and needs to be addressed.” The sentence needs some rephrasing to be in a passive form.
- L20. “we utilized 15 samples of three southern China local breeds (WS, LQ and 20 WN) to analyze the SNPs and identified some important immune related genes” The sentence needs some rephrasing to make clear what have been done in this study compared to reference 13 [Chen et al. 2018]. Previously published genomic sequences of Chinese local breeds have been used in order to explore genetic diversity and identify immune related genes.
- L26. “We report genome characterization of three native Chinese cattle breeds exploring ~34.3 M SNPs and ~3.8 M InDels using whole genome resequencing.” Would be “discovering” more appropriate than “exploring” here? It is not clear what have been done compared to reference 13 [Chen et al. 2018]. It seems authors performed their own SNP calling compared to reference 13. In this case “discovering” will be more appropriate.
- L29. “Gene ontology analysis revealed four immune responses related GO terms were over represented in all samples, while two immune signaling pathways were significantly 30 over-represented in WS cattle.” What is the reference used in the gene enrichment analysis? Bos taurus reference genome? The reference has to be indicated in order for the results to make sense
- L32. “We also found some other immune-related genes, e.g. PGLYRP2, 31 ROMO1, FYB2, CD46, TSC1 in the three Chinese zebu breeds, altogether.” Are these genes previously characterized in Bos Taurus and identified in Bos indicus by the authors? The sentence is not clear and needs rephrasing.
Introduction
- L39.The first paragraph of the introduction is confusing as it seems than authors are mixing domestication and natural selection. Besides, in the sentence “The artificial selection on cattle, e.g. Holstein and Beef master, led to produce higher milk/meat than local cattle breeds”, “genetic selection” will be more appropriate than “artificial selection”.
- L45. “and these cattle breeds constitute an important world heritage and a unique genome resource.” I suggest to use plural here (genomic resources) as there are several breeds.
Material and Methods
- L66. A brief reminder of sequencing methodology used by reference 13 to produce these sequences will be useful. (paired end library, insert size 500bp, Hiseq2000 Illumina platform, read length etc…).
- L70. I suggest indicating which is the species of the reference genome used in the mapping process for non-specialist readers (GCF_002263795.1, Bos Taurus, breed Hereford). Besides, it is not clear if authors performed their own read mapping or used mapping generated by reference 13. Parameters look similar as in reference 13.
- L88. “using ARS-UCD1.2 database” Genome annotation from Bos Taurus reference?
- L88. “For all the individuals, the variants were filtered as >5 SNPs/gene, whereas >5 InDels/gene were identified per breed [3].” Re-phrasing in order to make clear the reason for this filtering is clearly needed here. Are authors focusing on genes with non-synonymous polymorphisms as in reference 3 [Weldenegodguad et al. 2018]? It is very unclear why and what type of filtering have been done (all SNPs/indels, non-synonymous SNPs/indels?).
- L96. The reference used in the gene ontology analysis needs to be specified.
Results
- L111. “Compared with NCBI dbSNP bovine, a total of 9.33 million novel SNPs were identified, whereas, approximately 4.35, 5.29 and 5.23 million SNPs were previously identified and annotated in LQ, WN and WS cattle.” This sentence does not match table 2, as according to table 2, already known SNPs are 14.82, 17.80 and 18.20 million in LQ, WN and WS. What is considered novel/known? Is novel what was found by the authors and not by reference 13? Or is novel corresponding to SNPs that are not present in Bos Taurus breeds? One or two sentences are needed to ease understanding.
- In figure 2 (a), the sums of all number (4,040,822) is not 37 million, whereas authors claimed then identified 37 million SNP. What is the reason? Could it be possible than diagrams have been inverted between (a) and (b)?
- In table 4, both the counts for reference and the counts for the breeds of interest should be indicated. Besides, the reference should be indicated (Bos Taurus? breed Hereford?).
Discussion
- L181. “As for the heterozygous and homozygous ratio were detected SNPs, the higher ratio of these three breeds suggests that their population structure maybe normal and have a high heterozygosity rate.” This sentence is not clear and needs rephrasing.
- L191. “were private SNPs” What is the meaning? SNPs specific to this study?
- L196. “However, less than or equal to 3 bp occupies all InDels 83.07% which is comparable to the previous reports” This sentence needs some rephrasing.
- L197-L218. Are there any references or phenotypic data showing differential response to bacterial infection or inflammatory response in these three cattle breeds compared to Bos Taurus breeds in order to support genomics finding?
Reviewer 2 Report
Materials and methods
Lines 65-67: Despite the dataset utilized was previous described by Chen et al. (2018), a very brief description of the samples collection, DNA extraction, library preparation and sequencing should be provided for the readers.
Linha 91: remove an empty space.
Results
Lines 108-109: Considering that QualByDepth option was used normalizing quality by depth, minimum and maximum values for quality and depth of SNPs and INDELs after filtration, should be provided in the results.
Line 109: SNP density should be reported as SNPs/kb or SNPs/Mb as adopted in most of the articles published.
Lines 142-142: The sentence starting with “Disruptive inframe…” is confuse. Re-write.
Line 151: Why “filtered genes”? Filtered based on what? If the filtering was based on SNPs and INDELs and breeds, the authors should specify the gene lists utilized to perform GO enrichment, in Methods section.
Discussion
Lines 193-194: A little discussion about the differences observed in the number os variants between the breeds, should be included. Why is this difference observed? What can influence the number of variants?
Major comment: the authors should perform a spell check.
